# Enhancement Properties of Zr Modified Porous Clay Heterostructures for Adsorption of Basic-Blue 41 Dye: Equilibrium, Regeneration, and Single Batch Design Adsorber

**DOI:** 10.3390/ma15165567

**Published:** 2022-08-13

**Authors:** Saheed A. Popoola, Hmoud Al Dmour, Souad Rakass, Is Fatimah, Yan Liu, Ahmed Mohmoud, Fethi Kooli

**Affiliations:** 1Chemistry Department, Faculty of Science, Islamic University of Madinah, Madinah 42351, Saudi Arabia; 2Department of Physics, Faculty of Science, Mu’tah University, Mu’tah 6170, Jordan; 3Laboratory of Applied Organic Chemistry (LCOA), Chemistry Department, Faculty of Sciences and Techniques, Sidi Mohamed Ben Abdellah University, Imouzzer Road, P.O. Box 2202, Fez 30000, Morocco; 4Department of Chemistry, Faculty of Mathematics and Natural Sciences, Universitas Islam Indonesia, Kampus Terpadu UII, Jl. Kaliurang Km 14, Sleman, Yogyakarta 55584, Indonesia; 5Institute of Sustainability for Chemicals, Energy and Environment, 1 Pesek Road, Jurong Island, Singapore 627833, Singapore; 6Petroleum Technology, Operated Offshore Oil Field Development, Qatar Energy, Doha 3212, Qatar

**Keywords:** porous clay heterostructures, basic blue 41, acidity, adsorption, regeneration, single stage batch design

## Abstract

Zirconium porous clay heterostructures (Zr-PCH) were synthesized using intercalated clay minerals by zirconium species with different contents of zirconium. The presence of zirconium and silica species was confirmed by X-ray diffraction, X-ray fluorescence, and magic-angle spinning nuclear magnetic resonance. The insertion of zirconium improved the thermal stability, the specific surface area with a maximum of 950 m^2^/g, and the acidity concentration of 0.993 mol of protons per g of solid. These materials were used to adsorb the basic blue-41 from aqueous solution. The adsorption efficiency was examined at different conditions, with a maximum adsorbed amount of 346 mg/g as estimated from Langmuir model. This value was dependent on zirconium content in the PCHs. The adsorption process was found to be favorable and spontaneous. The efficiency of the spent materials was maintained after five reuse cycles with a decrease by 15% of the original value for a particular Zr-PCH material with a Zr content of 6.82%. Single stage batch adsorber was suggested using the mass balance equation and Langmuir isotherm model. The amount of PCH materials required depended on the target percentage of adsorption at specific volume and initial concentration of the basic-blue-41 dye solution.

## 1. Introduction

The porous clay heterostructures (assigned as PCHs) resulted from a combination of clay minerals hosts and mesoporous silica. Firstly, the interlayer spacing clay mineral was expanded by ammonium alkyl surfactants with chain length of 12 to 18 carbons. Secondly, a mixture of tetraethyl orthosilicate as a silica source (TEOS) and amine solution was reacted with the expanded organoclays at specific ratios [1]. In general, the resulting hybrid precursors were calcined to eliminate the surfactants and to obtain the PCH materials [1]. Another method to remove the surfactants was to treat the PCH hybrid precursors with ammonium salt solutions at mild temperatures [2]. The unique pore size distribution of PCH bridges the gap of pore-size region between microporous pillared clays (PILC) and mesoporous silica. The physico-chemical properties of PCH materials depended on the starting clay minerals and the length of the used surfactants [3,4]. The pore size and the acidity are the major parameters of interest for the specific applications of these materials. These properties can be tuned by the insertion of certain metals in the silica framework or by impregnation after the preparation of PCH materials [5,6,7,8,9,10,11]. The synthesis procedure requires different steps and generation of chemical waste and time consuming, thus affecting the economic and operational feasibility of these processes.

The PCH materials were considered as a hybrid precursor between organoclays and pillaring process. Starting from this idea, Kooli et al. (2014) came up with a new idea to reduce the chemical waste and the steps during the preparation of metal doped PCHs. [12]. It consisted of expanding the clay layers with polyoxocations of Al_13_ cations rather than organoclays as starting precursors. This method has allowed the direct insertion of Al species in the silica framework of the PCHs, as confirmed by different characterization techniques [12]. The proposed method has allowed to tune the contents of Al in the prepared PCH materials [13]. However, when pillared clay minerals were employed, the preparation of PCH precursors was not successfully achieved as indicated by XRD technique, due to cross linking between the pillaring species and the clay sheets, which made them difficult to be exchanged with the silica species and the amine surfactants [12]. Recently, using different method of synthesis, the PCH material could be prepared from iron pillared clay minerals using ultrasonic technique [14].

Taking into account the pore size of the PCH materials, they are used as molecular sieves to retain small molecules, or to capture volatile organic compounds [15,16,17,18]. In water purification, they have been used as adsorbents for phenolic compounds, heavy metals, anionic and cationic dyes, and pesticides [19,20,21,22,23,24,25,26]. In case of adsorption of volatile organic compounds, the hydrophobic surface character of PCH materials has favored a selective adsorption of certain VOCs. Meanwhile, the existence of silanol groups is responsible for alkaline metals’ adsorption which act similar to the cation exchange resins [27].

For the adsorption of phenol and chlorophenols, the presence of both hydrophobic and hydrophilic characteristics on the surface was related to silanol and siloxane groups that were formed during the pillaring and calcination of the PCH. These groups interact with the OH groups of phenol by hydrogen bonds [27].

The presence of silanol groups is a key factor to obtain high adsorption capacity. Thus, to enhance the proportion of silanol groups in PCH materials, the surfactants were removed using a softer process, such as treatment of ammonium chloride solution [28]. Another method to improve the adsorption and separation properties of PCH was to incorporate Zr or Al species in the silica framework during the synthetic step [29,30]. Indeed, the incorporation of Zr in the walls of PCH materials enhanced the dye adsorption capacity for the smaller anionic dyes, and the adsorption of cationic dyes took place by the sum of electrostatic and non-electrostatic interactions with the surface support between the silanol groups of the PCHs with the dye binding sites such as amine or hydroxyl [23,24]. In all mentioned works, only a fixed amount of Zr was incorporated to modify the adsorption and acidic properties of the resulting materials.

The innovation of this study is to modify the properties of Zr-PCH materials by direct insertion of different amounts of Zr in the silica framework of PCHs using intercalated clay precursors with different contents of Zr species. This method has helped to reduce the chemical waste and the steps during the synthesis process. In order to explore the properties of synthesized materials, X-ray diffraction (XRD), X-ray fluorescence (XRF), magic-angle spinning nuclear magnetic resonance (MAS-NMR), and nitrogen isotherms adsorption have been employed. These materials were assessed for the adsorption of Basic Blue-41 dye. Various parameters such as initial concentration, Zr content, solid concentration, pH of BB-41 solution, and temperature of adsorption were also examined.

The Basic Blue 41 has been selected as target dye due to its carcinogenic and mutagenic effects of this azo-dye [31]. The experimental data was used to assess the maximum adsorption capacity of the different Zr(X)-PCH materials. The reusability and regeneration are big concerns for the economic feasibility of the process. The regeneration has been evaluated for seven successive cycles using a friendly method to study the feasibility of these materials. Single batch design for the adsorption of Basic-Blue-41 was suggested using the Langmuir adsorption model.

## 2. Experimental Part

### 2.1. Materials

Raw clay (Ca-Mt) was acquired from source clays repository based in Purdue University. Zirconyl nitrate dihydrate (ZrO(NO_3_)_2_·2H_2_O)), tetraethylorthosilicate (TEOS), dodecylamine (C_12_H_27_N, C_12_ amine), cobalt nitrate (Co(NO_3_)_3_.6H_2_O), oxone (2KHSO_5_·KHSO_4_·K_2_SO_4_), cyclohexylamine (C_6_H_11_NH_2_) were used without further purification. Basic Blue-41 dye (C_19_H_23_N_4_O_2_S·CH_3_O_4_S) was provided by Aldrich. Deionized water was used during all the experiments.

### 2.2. Preparation of Zr-Intercalated Clays and Zr-PCH Precursors

Zirconyl nitrate dihydrate solution was used to prepare the Zr-intercalated clays, at four Zr mmol Zr/clay ratios (3, 6, 12, and 24). First, Zirconyl nitrate dehydrate solution (200 mL of 0.06 M) was aged at 80 °C for 1 h, and ageing of zirconium pillaring solution leads to enhance the degree of polymerization of zirconium species, due to an increase in the rate of hydrolysis. Then, 4 g of powder clay was added to the aged solution and stirred for an additional 1 h at 80 °C. The slurry was cooled and the solid was filtrated and frequently washed with deionized water [32]. The samples are identified as Zr(X)-Mt, where X represents the starting Zr-mmol to g of clay ratios.

The preparation of Zr doped PCH precursors was carried out by adding one g of each dried Zr(X)-Mt to a tetraethylorthosilicate (TEOS) and dodecylamine (C_12_H_25_NH_2_) corresponding to a molar ratio of clay/C_12_H_25_NH_2_/TEOS about 1/20/150. The admixture was completely mixed at room temperature for a duration of 4 h. The resulting material was filtrated and overnight air-dried [32]. The samples are designated as Zr(X)-PCH, i.e., Zr(3)-PCH will be identified as a PCH precursor using a Zr(3)-Mt intercalated clay.

The termination material will be used for Zr(X)-PCH precursors after calcination at 550 °C for a period of 4 h.

### 2.3. Adsorption Performance of Zr(X)-PCH

Batch adsorption experiments were performed to evaluate the adsorption performance of Zr(X)-PCH materials for BB-41 dye molecules. 0.050 g of solid was added to 50 mL BB-41 aqueous solution of initial concentrations (C_i_), ranging from 50 mg/L to 500 mg/L in sealed tubes without adjustment of solution pH. Experiments were conducted in duplicate in a water bath shaker at 120 rpm and at a constant temperature of 25 °C for overnight. The supernatant was filtered, and the concentration (C_e_) was estimated by U-visible spectrophotometer technique.

The effect of parameters such as initial concentration of BB-41, the solid concentration, pH of BB-41 solution, and adsorption temperature was studied. In order to obtain the optimum value for each parameter, it was varied while keeping the other parameters constant.

### 2.4. Regeneration and Reuse Experiments

The regeneration experiments were carried out using a mixture of cobalt nitrate and oxone solutions, as reported in previous studies [33]. Zr(3)-PCH and Zr(6)-PCH samples were treated first with a fresh solution of BB-41 (50 mL with C_i_ of 200 mg/L) for 8 h at room temperature. The fresh spent materials were separated by centrifugation and added to a solution of oxone and cobalt nitrate, then mixed for 30 min. The treated solids were added again to a BB-41 solution (50 mL with C_i_ of 200 mg/L) for 8 h, filtrated and analyzed with UV spectrophotometer to estimate the C_e_, and thus the adsorption efficiency percentage was determined. A similar procedure was carried out for the second cycle, and so on.

### 2.5. Characterization Techniques

X-ray fluorescence (XRF) instrument, from Bruker Model S4 was used to estimate the chemical composition of the Zr-clays and their PCH derivatives. Samples were dispersed in Boric acid and pressed in a form of disk. The success of Zr intercalation and the synthesis of Zr(X)-PCH materials were followed up by X-ray diffraction (XRD) technique. Bruker diffractometer (model Advance 8) equipped with a Ni filter and Cu-K alpha radiation (λ = 1.5406 Å) was utilized to collect the powder XRD patterns, in the range 2θ, from 1.5 to 50. ^29^Si solid magic-angle spinning nuclear magnetic resonance (^29^Si MAS NMR) spectra for the intercalated materials and related PCH materials were collected using a Bruker spectrometer as reported elsewhere and tetramethylsilane (TMS) as reference for the zero-chemical shift. To estimate the microtextural properties of the samples, nitrogen adsorption isotherms were collected using a Micromeritics apparatus model ASAP 2020. The samples were outgassed at 200 °C for overnight. The specific surface area (SSA) was estimated using the Brunauer–Emmett–Teller (BET) equation. The micropore volumes and surface areas were determined using the t-plot according to De Boer’s method. The total pore volume (TPV) of the studied materials was calculated from the amount of adsorbed nitrogen at a relative pressure of 0.97.

The amount of acid sites was estimated using the cyclohexylamine probe molecule. The samples were first saturated with a cyclohexylamine solution for overnight, and dried at 80 °C in air to desorb the physiosorbed cyclohexylamine molecules. The mass loss between 290 °C and 420 °C was calculated to approximate the acidity in relation of mmol of cyclohexylamine per g of sample [13]. The thermogravimetric analysis was performed using a TA model 9610 (TA instruments, New Castle, DE, USA) from 25 °C to 600 °C at a rate of 5 °C/min, under air. As mentioned in Section 2.3. The concentrations of BB-41 were estimated using Ultraviolet-Visible (UV-Visible) spectrophotometer, from Varian model carry 100 at the absorbance at maximum wavelength (λ_max_ = 610 nm).

## 3. Results and Discussion

### 3.1. Characterization of Zr(X)-PCH Materials

The PXRD patterns of Zr(X)-Mt intercalated clays are presented in Figure 1. An increase in the d_001_ values from 1.51 nm (Figure 1 (a)) to 2.24 nm was associated with the intercalation of zirconium polyhydroxy cations into the interlayer spacing of the clay mineral, and it depended on the used mmol of Zr/clay ratios; it varied from 1.91 nm for Zr(3)-Mt (Figure 1 (b)) to 2.24 nm for the Zr(24)-Mt (Figure 1 (e)). The resulted expansion of interlayer space could be assigned to a double-layer flat-lying Zr tetramers or to a single layer of Zr tetramer perpendicular to the clay layers [32,34].

When Zr(X)-Mt precursors were reacted with TEOS and C_12_ amine, the resulting Zr(X)-PCH precursors exhibited PXRD patterns with an intense reflection corresponding to d_001_ values of 3.80 (Figure 2 (b)) to 4.10 nm (Figure 2 (e)), and no multiple reflections were observed, consistent with previous reports for conventional PCH ones. The pre-expansion of the interlayer space with Zr species, was a mandatory step to successful synthesis of PCH precursor. Indeed, when raw clay (without intercalation of Zr species) was reacted with TEOS and C_12_ amine, the resulting PXRD pattern of Zr(0)-PCH precursor was different and exhibited a interlayer space of 1.71 nm (Figure 2 (a)), corresponding to intercalated clay with different silica species [35]. In the case of zirconium PCH prepared by different method, the PXRD pattern did not exhibit 001 reflection and other basal reflections, due to a random delamination of the clay sheets upon insertion of ZrO_2_-SiO_2_ species [36].

After thermal treatment at 550 °C, the d-values of the precursors shifted to higher angles, due to the loss of C_12_ surfactants and condensation of silica species via dihydroxylation. A maximum value of 3.81 nm was recorded for Zr(6)-PCH material (not shown). These values were comparable to those reported for PCH ones doped with variables Al contents [12]. In the case of Zr(0)-PCH precursor, the d-value continued to decrease to a value 1.14 nm. This value was larger than the value of raw clay calcined at the same temperature (0.96 nm), due the existence of some silica pillars within the interlayer space of Mt clay.

Figure 3 depicts the PXRD patterns of Zr(6)-PCH precursor heat treated at different temperatures. Calcination at 550 °C, the value of the d_001_ value did not change, however, at 650 °C, the intensity of the 001 reflection was drastically reduced and the PXRD patterns exhibited broad reflection embedded in the background of the patterns. This fact could be linked to the dehydroxylation of the clay layers host and destruction of the mesoporous silica framework or to the delaminated structure of the calcined materials [36]. The reflection related ZrO_2_ (in tetragonal) phase was not detected (Figure 3 (c)) and could indicate that ZrO_2_ species were incorporated in the silica framework.

The data of XRF analysis supported the PXRD data and confirmed the presence of zirconium and silica in different precursors (Table 1). The content of ZrO_2_ percentage increased with initial Zr mol/clay ratios, with an uppermost value of 16.70% for Zr(24)-Mt. The decrease in CaO content was the result of the complete exchange Ca^2+^ cations during the modification process. A decrease of Al_2_O_3_ and MgO contents also occurred.

An increase of SiO_2_ percentage was observed for Zr(X)-PCH precursors, and as expected, the values for the other major elements decreased compared to Zr(X)-Mt precursors. Mainly, the percentage of ZrO_2_ was reduced from 8.12% to 4.21%. The proposed method has allowed the modification of the Zr contents in the Zr(X)-PCH materials.

The evolution or rearrangement of silicon species in Zr containing clay derived materials was investigated by ^29^Si MAS-NMR technique. The ^29^Si MAS-NMR spectra of Zr(X)-PCH precursors exhibited similar spectra, the one of Zr(6)-PCH was presented as an example (Figure 4). Additional resonances at −100.4 ppm and −109.7 ppm were observed, which are attributed to Si(OSi)_3_OH (Q^3^) sites and Si(Osi)_4_ (Q^4^) centers in the developing silica framework, respectively [37,38]. Compared to Zr(6)-Mt precursor (Figure 4 (a)), where mainly an intense band at −95.6 ppm is related to the Q^3^ Si atoms in the tetrahedral layer of the clay, linked through oxygen to three other Si atoms and either to an Al or Mg atom in the octahedral layer [37], while the broad band at −111.8 ppm was assigned to the silica impurities in the starting raw clay. When Zr(6)-PCH precursor was calcined at 550 °C, a wideness of the resonance band at −108.1 ppm was observed, due to the dehydroxylation and condensation of silica species between the clay layers (Figure 4 (c)). The cross-linking processes occurred during the thermal treatment, resulting in improvement of the Q^4^ resonance relative to other components of the NMR signals [37]. In addition to Q^3^ mesoporous silica at −100.4 ppm, the resonance band at −95.6 ppm (associated to the parent clay) was reduced in intensity, and it was embedded in the intense band of the silica varieties. These data are in agreement with PXRD data, and implied that the structure of Zr(X)-PCH material was preserved with some conversion of certain silica-inserted species.

When Zr-PCH(6) precursor was treated at temperatures higher than 650 °C (Figure 4 (d)), the spectrum exhibited mainly a broad resonance band around −108.1 ppm, associated to Q^4^ Si species, the intensity of the band at −100.4 ppm reduced with calcination temperatures, and could indicate the destruction of the silanol groups. The broadness of the resonance bands was the result of coexistence of different silicon species mixed in one phase. At 850 °C, only a broad band at −108.1 ppm was detected (Figure 4 (f)).

The Zr(X)-PCH materials (calcined at 550 °C) displayed larger specific surface area (SSA) values ranging from 445 to 950 m^2^/g than the starting Zr(X)-Mt precursors (from 231 to 318 m^2^/g) and varied according to mol of Zr/g of clay ratios. Indeed, the Zr(0)-PCH exhibited the lowest SSA value of 180 m^2^/g. The highest value of 950 m^2^/g was recorded for Zr(6)-PCH, and continued to reduce as the mol Zr/g of clay ratio increased. Interestingly, Zr(X)-PCH materials exhibited some microporous character, in addition to the main mesoporous one, as indicated by average pore diameter and pore volume values. Compared to PCH materials synthesized by conventional route, the insertion of Zr species in PCH materials has improved the microtextural properties [13,36] (Table 2).

The microtextural values of Zr(6)-PCH precursor also depended on the calcination temperatures. For example, Zr(6)-PCH calcined at 650 °C, a slight reduction was observed, and it continued to decrease with further increase of calcination temperatures (Table 3). A reduction of three folders was noticed at 900 °C for specific surface area value, with a complete vanish of the micropore contribution.

The acidity of the different Zr(x)-PCH materials was expressed in mmol of protons (H^+^)/g of the sample, as reported in Section 2.5.

The introduction of Zr species generates acid centers compared to material prepared from raw clay (without insertion of Zr species), and the estimated acidity decreased with the increase of the Zr content in Zr(X)–PCH materials (Table 3). The acidity of Zr(x)-Mt pillared materials exhibited lower values in three less folders compared to the Zr(X)-PCH derivatives.

### 3.2. Parametric Studies of Adsorption of Basic Blue-41

#### 3.2.1. The Impact of Initial Concentration

The adsorption of BB-41 by the surface of the solid was strongly dependent on the initial concentrations (C_i_) values. Figure 5 depicts the behavior of Zr(12)-PCH material. An increase in the initial concentration (C_i_) from 50 mL/g to 250 mg/L leads the adsorption capacity to increase with a sharp slope from 77 mg of BB-41/g of solid to 265 mg of BB-41/g of solid. However, for C_i_ values above 300 mg/L, the adsorption slope decreased until it reached an almost constant value at C_i_ values higher than 300 mg/L. The maximum experimental adsorption capacity was 285 mg of BB-41/g of solid.

On the other hand, the percentage adsorption efficiency (R%) of BB-41 continued to decrease with C_i_ of BB-41 dye from 83% to 28% (Figure 5). In other words, the residual concentration of BB-41 molecules will be higher for higher C_i_ values. The higher adsorption amounts may be related to the availability vacant adsorption sites available at initial stage of the adsorption process, resulting in an increased concentration gradient between the solute in solution and the solute on the adsorbent surface. However, at high C_i_ values the available sites of adsorption decrease, thus the adsorption percentage of BB-41 is reduced [23]. In all the cases, the modification of the raw clay was beneficial to improve its removal properties.

#### 3.2.2. Effect of Zr Contents

The adsorption of BB-41 on different Zr(X)-PCH materials have nearly the same shape with similar characteristics. The Zr species being introduced in the PCH materials increased the adsorption properties. The adsorbed amounts were strongly dependent on the Zr content in the PCH materials (Figure 6). Maximum experimental adsorption of 310 mg/g was achieved for Zr(6)-PCH material, and a minimum of 240 mg/g was obtained for Zr(3)-PCH one. Zr(12)-PCH and Zr(24)-PCH materials exhibited an important adsorption capacities within intermediate values. The decrease of the adsorption BB-41 indicated that some Zr species were not available to the adsorption sites of BB-41. Perhaps this availability is hampered by change in the crystallinity of Zr species with an increase in the content of Zr in PCH materials, or to the blocking of pores by the BB-41 dye molecules [39].

#### 3.2.3. Effect of Solid Concentration

In this study, the C_i_ is fixed to 200 mg/L and different masses of Zr(12)-PCH were added to 50 mL of this solution for overnight. The results showed that adsorption percentage (R%) of BB-41 was improved with an increase in Zr(12)-PCH concentration (g of sample/volume of BB-41 solution (L), Figure 7). The R% was improved from 26% up to 75% by varying the solid concentration from 0.20 g/L to 4 g/L. In the case of Zr(3)-PCH and Zr(24)-PCH, similar trends were noticed, however, the adsorption percentage (R%) values were lower, and additional dosages were requested to achieve the same R% of Zr(12)-PCH. However, the removed amount of BB-41 (mg/g of of Zr(X)-PCH material) decreased with increasing the solid concentrations. The R% of 100% was attained for Zr(6)-PCH material as a solid concentration of 8 g/L.

The surface concentration (*Γ*) of the adsorbed BB-41 (mass (mg) per one m^2^ of surface area) was determined by Equation (1) [40]
*Γ* = V(C_i_ − C_eq_)/*m*_s_*a*_s_(1)
where C_i_ is initial concentration, C_e_ is equilibrium concentration, *m*_s_ is the mass of used Zr(12)-PCH with a specific surface area of *a*_s_, and V is the volume of BB-41 solution. When data are stated in terms of surface concentration *Γ* (mg/m^2^), the surface concentration decreased from 0.28 to 0.01 as the mass of Zr(12)-PCH material increased, indicating that the available surface area and the adsorption sites were not fully covered by the present BB-41 molecules (Figure 7).

#### 3.2.4. Effect of BB-41 pH Solution

One of the largest factors that affects the adsorption of liquid onto solid surface is the pH. Indeed, the pH influences the solid charge, the degree of ionization, and the dissociation of the functional groups of the active sites of the adsorbent, in addition to the ionization of dye molecules [41,42]. The influence of pH on the BB-41 adsorption by Zr(12)-PCH materials was studied in the pH range of 2 to 12 for a C_i_ solution of 200 mg/L and by adding a few drops of HCl (0.1 M) or NaOH (0.1 M) solutions.

In general, the adsorption percentage (R%) was enhanced with the increase of pH up to 6, then it reached a maximum of 82% at pH values between 7 to 9. However, using Zr(6)-PCH material, an adsorption percentage reached 100% at pH higher than 6 (Figure 8). Regarding the other tested Zr(X)-PCH materials, the maximum of 100% was not attained in the same range of 7 to 9. At pH values higher than 10, the BB-41 molecules decomposed in strong alkaline environment and formed a brown precipitate [33]. Thus, the analysis of supernatant could not be followed up.

The point of zero charge pH (pH_pzc_) is the pH at which the zero electrical charge of the adsorbent surface is achieved [42]. At pH < pH_pzc_, a positive surface charge dominates, and Zr(X)-PCH materials exhibited a net positive charge. Repulsion between same charge occurred and partial adsorption occurred. The pH_pzc_ of the raw clay found to be in the range between 8 and 9.5 and close to 7.5 [43], and it shifted to acidic values of 4.1 to 4.7, after pillaring process. This shift was due to the acidic character of the zirconium species used in the pillaring process [34,44]. The pH_pzc_ of Zr(X)-PCH materials continued to diminish to reported values in the range of 3 to 4 [26]. At pH > pH_pzc_, Zr(X)-PCH materials exhibited a net negative charge due to deprotonation of the silanol groups (Si-OH) in the silica framework species intercalated in the prepared Zr(X)-PCH materials. Indeed, the presence of these groups has been reported on the surface of PCH [28,37]. Thus, at higher pH values, the adsorption of BB-41 was significant.

#### 3.2.5. Isotherm Modelling

Previous studies have indicated that the Langmuir model was the most adequate to estimate the maximum adsorption capacity (q_max_) loaded on the surface of a solid [45]. This model supposes that one monolayer of adsorbate is formed on the solid surface. The linear model of Langmuir model is presented in Equation (2).
(2)$$\frac{C_{e}}{q_{e}}=\frac{1}{q_{\max}\cdot K_{L}}+\frac{C_{e}}{q_{\max}}$$

This relates the adsorbed amount of BB-41 (q*_e_*) to the concentration at equilibrium (C*_e_*). q_max_ is the maximum adsorption capacity of BB-41 required to cover the surface by a complete monolayer (mg of BB-41/g of solid), and K_L_ (L/g) expresses the Langmuir constant. The deduced Langmuir parameters for the used samples are displayed in Table 4.

The raw clay displayed a maximum adsorption capacity (q_max_) of 57 mg/g close to that described for similar materials [46]. This value was enhanced and depended on the ZrO_2_ content, the highest value of q_max_ (114 mg/g) was achieved for Zr(6)-Mt, and the lowest value was reported for Zr(24)-Mt material. The improvement of the adsorption capacity of Zr(X)-Mt samples was assigned to the presence of Zr species. In the case of Zr(X)-PCH derivatives, continuous enhancement of the adsorption capacity of 2 to 3 orders of magnitude was obtained. The highest q_max_ of 346 mg/g was achieved for Zr(6)-PCH, and a lowest value of 224 mg/g was obtained for Zr(3)-PCH material.

The *q*_max_ decrease of Zr(X)-PCH with Zr higher contents indicated that some Zr species could block the pores of the PCH materials. Indeed, there is a general trend in the decrease of average pore diameters (APD) values of Zr(X)-PCH materials with the increase of the Zr/Si contents.

The maximum adsorption capacity (q_max_) values (mg/g) were transformed to maximum surface concentrations (*Γ*_max_, mg/m^2^), as described in Equation (1) and reported in Table 4. The maximum surface concentration attained a maximum of 0.505 mg/m^2^ for Zr(3)-PCH material in the actual experimental conditions. Next, it was decreased as the content of Zr increased in PCH materials, and it reached a value of 0.292 mg/m^2^ for Zr(24)-PCH material. A three-dimensional calculation indicates that the BB-41 dye has a planar structure with the following dimensions: 1.716 nm (length), 0.665 nm (width), and 0.665 nm (thickness). These values were close to that reported in the literature [47]. From the planar area of BB-41, the monolayer capacity was estimated to 0.982 mg/g. The results indicated that q_max_ values for all the used materials did not exceed a monolayer of BB-41 molecules under the present experimental conditions.

Collection of adsorbents used in the literature for the adsorption of BB-41 is reported in Table 5. Only the silica and aluminosilicates were mentioned in this present table. The Zr(X)-PCH materials exhibited noticeable maximum adsorption capacities (q_max_) values. The difference could be related to the presence of Zr species in the silica framework, the mesoporous character of the Zr(X)-PCH materials, and their higher specific surface areas.

#### 3.2.6. Proposed Mechanism of Adsorption

The optimal conditions to get good performance of Zr(X)-PCH materials are starting Zr mmol Zr/clay ratio of 6, the pH values of BB-41 in the range of 7–9, solid concentration of 8g/L, and initial concentrations of BB-41 below 300 mg/L.

Basic-Blue-41 is known as cationic dye because the presence of positive charge in the dye molecule once is soluble in water. Basic dyes possess cationic functional groups such as amino (−NR_3_^+^) or dialkylaminogroups NR_2_^+^). The positive charge is not confined to a certain group but is distributed over the whole molecule. Indeed, the positive charges were situated on the nitrogen, sulfur atoms, and on the hydrogen atom of the hydroxyl group (OH) [23]. The maximum adsorption occurred within the range pH 7–9. In this pH range, the surface of Zr(X)-PCH materials is negatively charged (pH greater than the pH_zpc_) and the dyes are positively charged. The electrostatic force of attraction comes into play to increase the adsorption process due to the opposite charge on Zr(X)-PCH materials and dye molecule.

The adsorption of dye molecules could occur via the diffusion into the pores of Zr(X)-PCH materials. This process was assumed for other materials [51].

The solute molecules diffuse in pores structure of the adsorbents with diameter of 1.3–1.8 times higher than their diameters [51]. Based on the dimensions of BB-41 molecules and the pore of Zr(X)-PCH materials, the BB-41 molecules diffuse easily in these pores.

The actual pores occupation by BB-41 dyes could be estimated from the calculated volume of the adsorbed BB-41 molecules and the total pore volume of the Zr(X)-PCH materials. From the reported dimensions of BB-41 and maximum adsorbed amounts (*q*_max_) mentioned in Table 3, the estimated volumes of Zr(X)-PCH pores varied from 0.233 cm^3^/g to 0.360 cm^3^/g, corresponding an estimated pores occupation in the range of 37% to 59% (Table 6).

#### 3.2.7. Adsorption Thermodynamics

We explored the effect of temperature on the adsorption of BB-41 dye by Zr(6)-PCH material. The adsorption of BB-41 was carried out at four different temperatures: 25 °C, 30 °C, 40 °C, and 50 °C, using a C_i_ of 200 mg/L and solid concentration of 1 g/L, for overnight. The obtained results revealed that the adsorbed amount of BB-41 increased with the increase of temperature. The thermodynamic parameters of the adsorption reaction, ΔG° (standard free energy change, kJ mol^−1^), ΔS° (standard entropy, kJ mol^−1^ K), and ΔH° (standard enthalpy, kJ mol^−1^), were computed from the experimental results using the following Equations (3) and (4) [52]:ΔG° = −RTlnq_e_/C_e_(3)
ΔG° = ΔS°/R − ΔH°/RT(4)
where q_e_/C_e_ is the thermodynamic equilibrium constant, T is the adsorption temperature (K), and R is the universal gas constant (8.314 J mol^−1^ K). ΔG° was calculated from Equation (3) in the temperature range studied. The values of ΔH° and ΔS° were determined from the slope and intercept plot of ΔG° versus 1/T, as presented in Equation (4). The computed parameters are summarized in Table 7.

The positive value of ΔH° confirms that the adsorption process is endothermic, in consistent with the increase of the adsorbed amount with temperature values. The positive value of ΔS° accompanying the adsorption of BB-41 reveals the increased randomness at the solid–solution interface during the fixation of the adsorbed BB-41 cations on the surface of the sorbent. This observation suggests that the spontaneous adsorption of BB-41onto Zr(6)-PCH is controlled by entropy change [53]. The negative values of ΔG° indicate that the adsorption process is favorable and spontaneous thermodynamically. The absolute values of ΔG° increase with temperature, which suggests that the adsorption of BB-41 is more feasible and favorable at higher temperatures. Based on ΔG° and ΔH° values, it can be postulated that the adsorption of BB-41 is a physical adsorption process [53,54].

## 4. Zr(X)-PCH Reusability

The reusability of a spent adsorbent is an important factor from economic and environmental points of view [55]. The regeneration studies indicated that the removal of BB-41 for Zr(6)-PCH material remained steady after two cycles and then declined from 83% to 70% in the fifth run. Overall, a decrease of 27% of the starting value was noted after the seventh cycle of reuse. However, for Zr(3)-PCH material, the initial removal percentage (of 66%) was maintained only for one reuse cycle, and continuously declined to a value of 56% after three cycles. Reduction of 40% was attained after five cycles of reuse. (Figure 9). The Zr(24)-PCH behave in a similar manner than Zr(3)-PCH, where a reduction of the original value from 56% to 37% was achieved in the fifth run of reuse.

Overall, Zr(6)-PCH was sufficiently stable compared to the other used materials, and exhibited a good reusability without a significant loss of its removal efficiency.

The slight decrease of removal efficiency can be attributed to the easy destruction of the removed BB-41 molecules, which made the adsorption sites accessible to the successive adsorption runs, as reported in previous work [23]. In the case of significant decrease of removal efficiency, it could imply that the dye molecules were strongly bonded to the adsorption sites of the Zr(X)-PCHs. Moreover, the acidity of the Zr(X)-PCH material could activate the decomposition of the adsorbed BB-41 dyes by the used method [56]. Indeed, the Zr(6)-PCH material exhibited the highest acidity value as estimated by cyclohexylamine molecule (see Table 2).

## 5. Batch Design from Langmuir Model

Adsorption isotherms were applied in the single-stage batch adsorption design [57]. The objective was to predict the necessary mass (m (g)) of Zr(X)-PCHs to reduce an initial concentration of BB-41 dye from C_o_ (mg/L) to a specific concentration (C_1_) in a specific volume (V, (L)) [58], with a variation of adsorbed amount BB-41 (mg/g of Zr(X)-PCH) from q_o_ (mg/g) to q_1_ (mg/g). The mass balance equation at equilibrium condition is given as:(5)$$\left(C_{o}-C_{e}\right)=m\left(q_{o}-q_{e}\right)={\mathrm{mq}}_{e}$$

In the current investigation, the adsorbed BB-41 data fitted the Langmuir isotherm model well. The corresponding equation of the model can be replaced in Equation (5), and the reorganized form is displayed in Equation (6) [23].
(6)$$\frac{m}{V}=\frac{C_{o}-C_{e}}{q_{e}}=\frac{C_{o}-C_{e}}{\frac{q_{m}K_{L}C_{e}}{1+K_{L}C_{e}}}$$

Replacing C_o_ − C_e_ and C_e_ expressions by C_o_, Equation (6) could be rewritten as Equation (7):(7)$$\frac{m}{V}=\frac{C_{o}-C_{e}}{q_{e}}=\frac{{\mathrm{RC}}_{o}}{q_{m}\frac{K_{L}\left(1-R\right)C_{o}}{1+K_{L}\left(1-R\right)C_{o})}}$$

Figure 10 illustrate the plots obtained from Equation (7) to predict the required amount of Zr(3)-PCH (g) (Figure 10A) and Zr(6)-PCH(g) (Figure 10B), respectively, to reduce the C_i_ of 200 mg/L to 50, 60, 70, 80, and 90%, from different volumes of BB-41 in the range of 1 to 12 L in 1 L increment.

In general, the needed mass of Zr(X)-PCH materials increased with the raise of used volumes of the pollutant and desired adsorption percentage values. The reduction of a specified volume of 10 L with a C_i_ of 200 mg/L) to 50%, 60%, 70%, 80%, and 90% required, 3.0 g, 3.6 g, 4.4 g, 5.2 g, and 6.4 g of Zr(6)-PCH material. Nevertheless, the predicted amounts of Zr(3)-PCH were 3.5 g, 7.0 g, 8.9 g, 11.6 g, and 16.1 g, respectively, using the same conditions.

The required amounts of Zr(X)-PCH materials were lower using Zr(6)-PCH compared to the other studied materials. This difference was correlated to the higher adsorption efficiency of Zr(6)-PCH material for the adsorption of BB-41.

## 6. Conclusions

Zirconium-intercalated clay minerals were used to prepare Zr modified PCHs. The pre insertion of Zr species was an essential step in the synthesis of these materials. The zirconium contents in the modified precursors and the calcination temperatures have affected the physicochemical properties of Zr-PCH materials with high specific surface area value of 950 m^2^/g and large pore volume of 0.801 cc/g. The presence of Zr species improved the acidity of the PCH materials. The adsorbed amount of BB-41 improved with the presence of Zr species, higher pH of the BB-41 solution and initial concentrations, with a maximum of 346 mg/g achieved for Zr(6)-PCH material. Meanwhile, it decreased for 273 mg/g for Zr(24)-PCH material. In addition to the electrostatic interaction between the silanol groups and the cationic BB-41 dye, the adsorption of BB-41 molecules occurred through diffusion in the pores of Zr-PCH materials. The adsorption of BB-41 was found to be endothermic and spontaneous on Zr(6)-PCH material. The regeneration tests indicated that these materials could be used for at least five runs, starting from an initial concentration of 200 mg/L, with a loss of 15% of its efficiency. The design of single stage batch was suggested using Langmuir model, and 5.2 g of Zr(6)-PCH material were needed to reduce an initial concentration of 200 mg/L by 80%. The required mass increased with the aimed percentage adsorption and content of Zr in the PCH materials.

## Figures and Tables

**Figure 1 materials-15-05567-f001:**
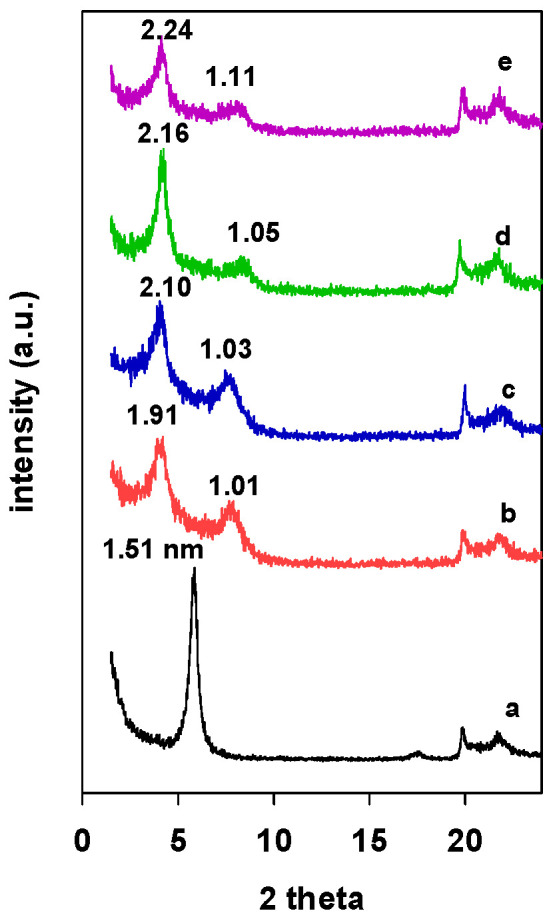
PXRD patterns of Zr intercalated clays (Zr(X)-Mt) with different starting Zr/clay ratios. (a) 0, (b) 3, (c) 6, (d) 12, and (e) 24.

**Figure 2 materials-15-05567-f002:**
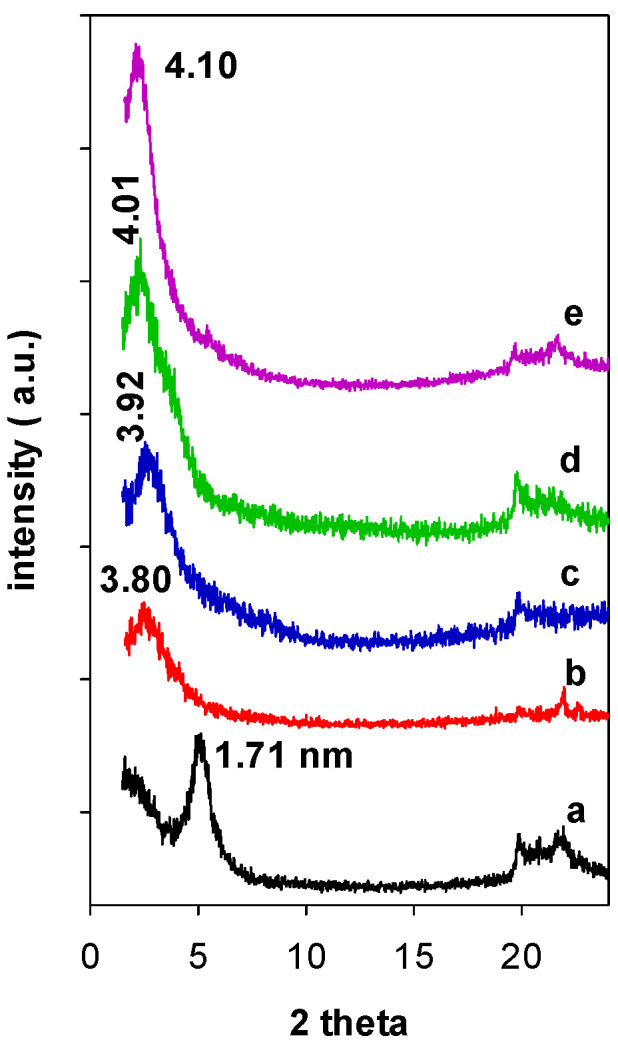
PXRD patterns of the Zr(X)-Mt after reaction with TEOS solution and C12 amine (a) 0, (b) 3, (c) 6, (d) 12, and (e) 24.

**Figure 3 materials-15-05567-f003:**
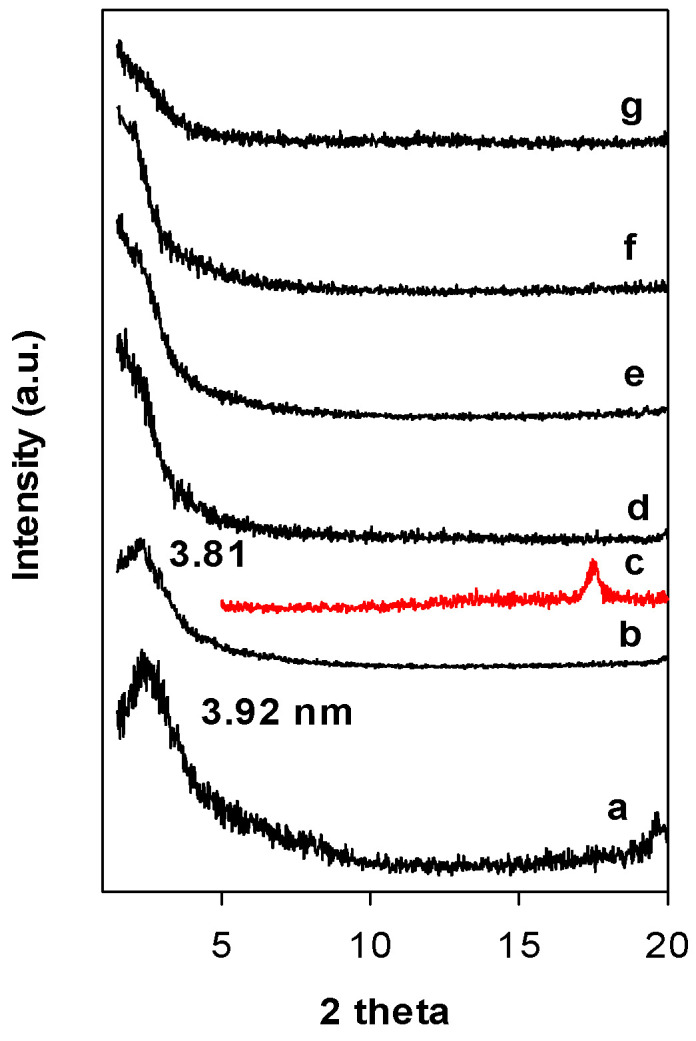
PXRD patterns of the Zr(6)-PCH precursor (a) calcined at different temperatures (b) 550 °C, (d) 650 °C, (e) 750 °C, (f) 850 °C, and (g) 900 °C. (c) Corresponds to tetragonal ZrO_2_ phase.

**Figure 4 materials-15-05567-f004:**
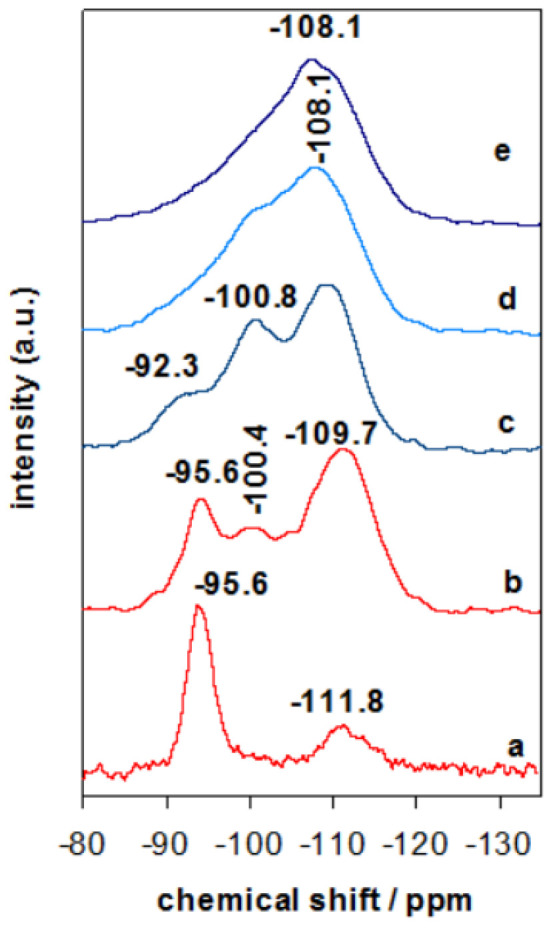
^29^Si-MAS-NMR spectra of (a) Zr(6)-Mt, (b) Zr(6)-PCH precursor calcined at (c) 550 °C, (d) 650 °C, and (e) 900 °C.

**Figure 5 materials-15-05567-f005:**
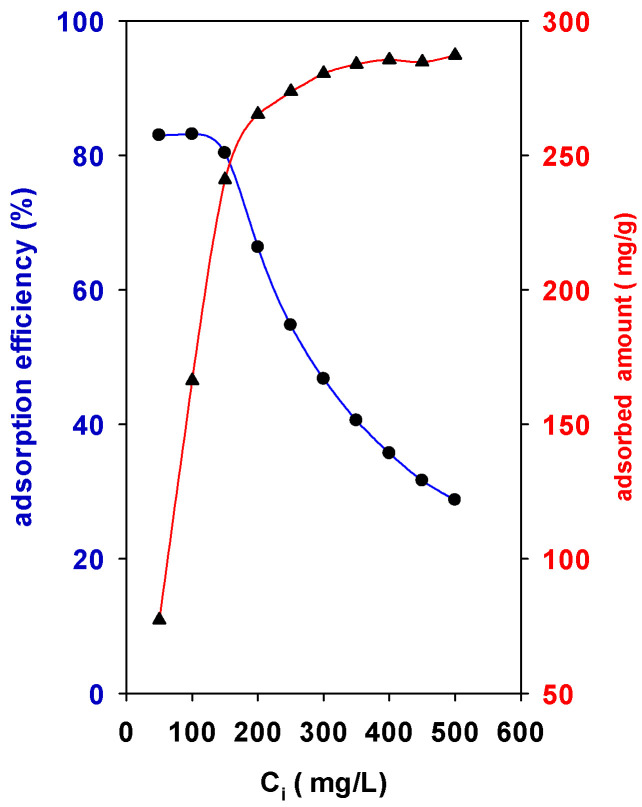
Effect of initial concentration (C_i_) on the (●) adsorption efficiency (%) and (▲) adsorbed amount of BB-41 (mg/g) using Zr(12)-PCH material.

**Figure 6 materials-15-05567-f006:**
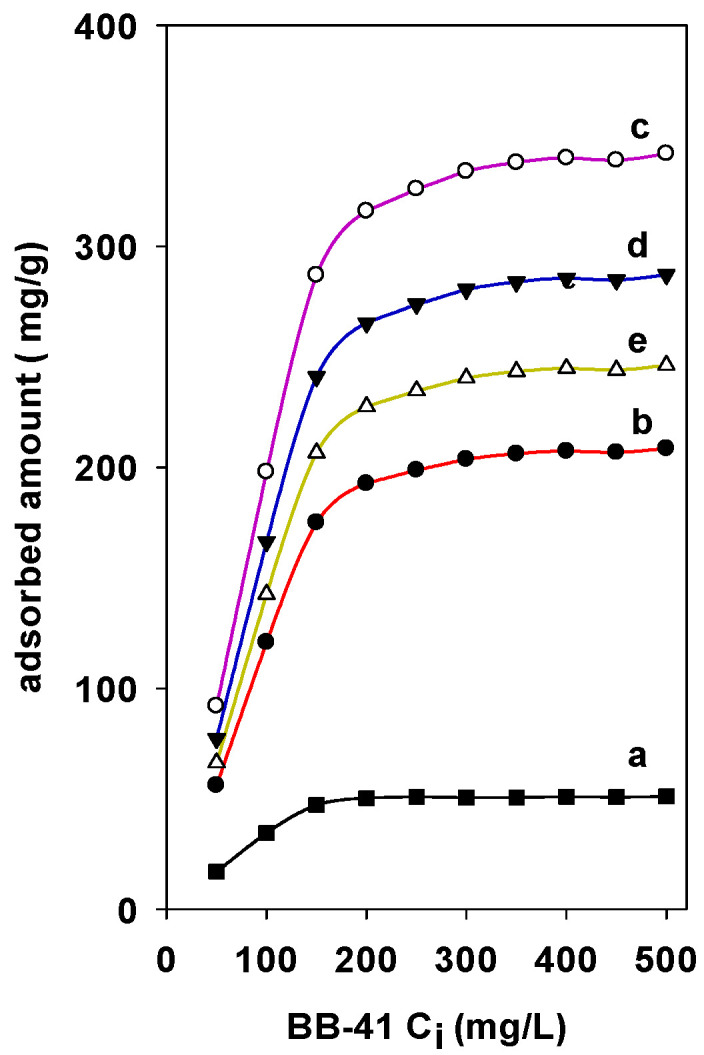
Effect of Zr contents in the Zr(X)-PCH materials on the adsorbed amounts of BB-41 dye. (a) Zr(0)-PCH, (b) Zr(3)-PCH, (c) Zr(6)-PCH, (d) Zr(12)-PCH, and (e) Zr(24)-PCH.

**Figure 7 materials-15-05567-f007:**
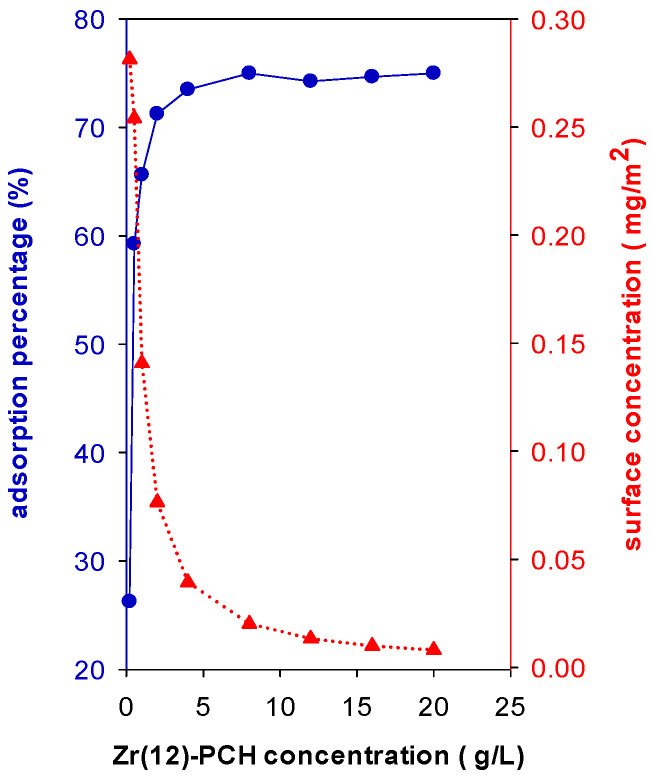
Effect of Zr(12)-PCH concentration (g/L) on the adsorption percentage of (R%) (●) and on the surface concentration (mg/m^2^) (▲) of Basic Blue-41 adsorption.

**Figure 8 materials-15-05567-f008:**
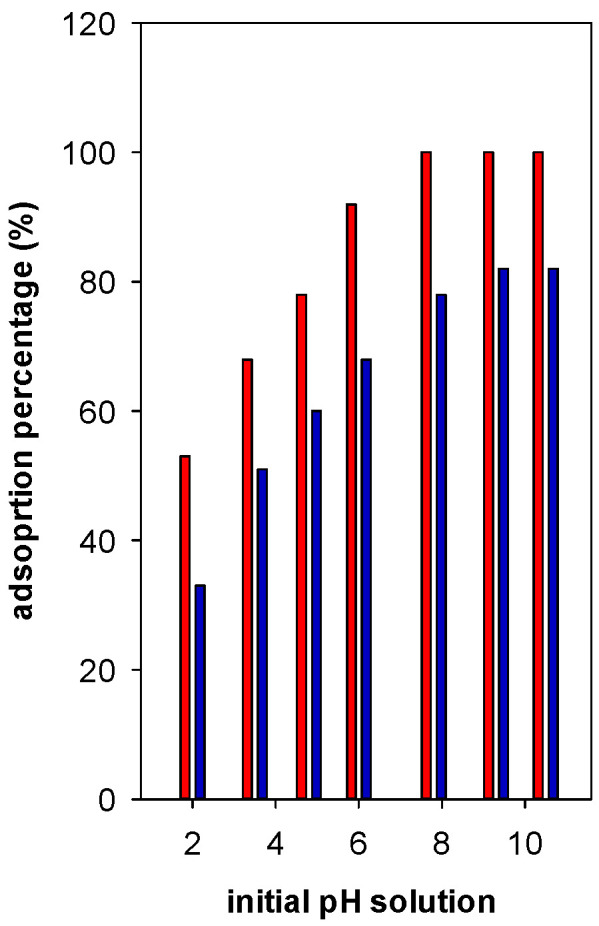
Effect of basic Blue -41 pH solution on the adsorption percentage (%) using Zr(6)-PCH (red bar) and Zr(12)-PCH (blue part) materials.

**Figure 9 materials-15-05567-f009:**
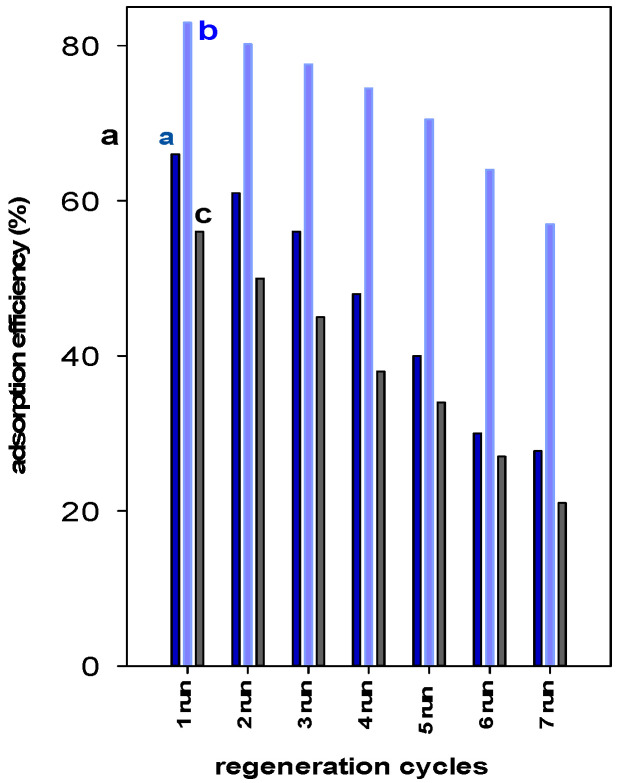
Reusability data of (a) Zr(3)-PCH, (b) Zr(6)-PCH, and (c) Zr(12)-PCH materials using a BB-41 initial solution of 200 mg/L.

**Figure 10 materials-15-05567-f010:**
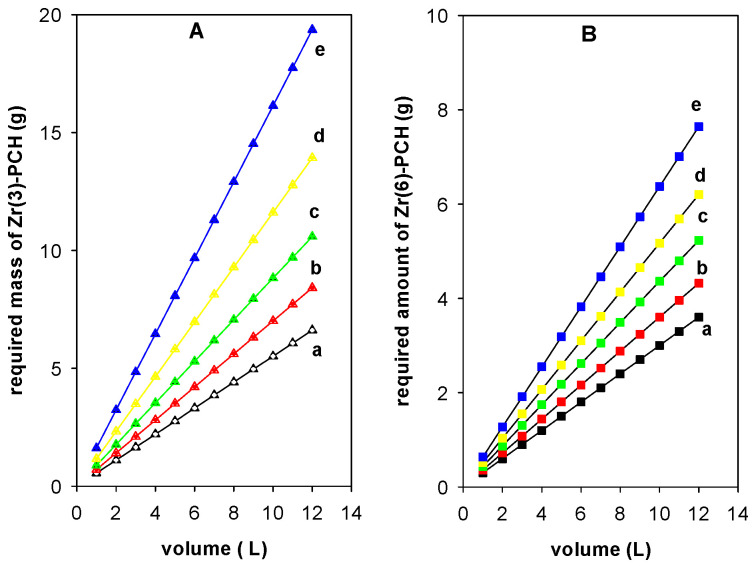
Prediction of the required amount (g) of (Zr(3)-PCH material (**A**) and Zr(6)-PCH (**B**) necessary to reduce BB-41 dye with C_i_ (200 mg/L) to (a) 50%, (b) 60%, (c) 70%, (d) 80%, and (e) 90%.

**Table 1 materials-15-05567-t001:** XRF analysis of Zr(X)-Mt precursors and Zr(X)-PCH derivatives.

Samples	SiO_2_	Al_2_O_3_	MgO	ZrO_2_	CaO	Zr/Clay +
Zr(0)-Mt	66.4	14.5	2.97	0	1.66	0
Zr(0)-PCH	72.33	8.5	1.53	0	0.91	0
Zr(3)-Mt	54.5	11.7	2.03	6.23	0.03	0.57
Zr(3)-PCH	78.52	5.53	1.02	4.21	0.02	N.C.
Zr(6)-Mt	49.1	10.4	1.9	11	0.02	1.12
Zr(6)-PCH	81.02	5.54	0.66	6.83	0.02	N.C.
Zr(12)-Mt	46.7	10	1.69	12.8	0.01	1.38
Zr(12)-PCH	82.21	5.21	0.62	7.25	0.02	N.C.
Zr(24)-Mt	46.4	9.7	1.65	16.7	0	1.79
Zr(24)-PCH	83.01	4.69	0.58	8.12	0.02	N.C.

Values between brackets correspond to Zr(X)-PCH materials. + experimental values (Zr mmol/g of clay). N.C. not calculated.

**Table 2 materials-15-05567-t002:** Microtextural properties and acidity of Zr(X)-Mt and their Zr(X)-PCH derivatives.

Samples	SSA (m^2^/g)	µpore Volume (mL/g)	µpore SA (m^2^/g)	TPV (cm^3^/g)	APD (nm)	Acidity mmol H^+^/g
Zr(0)-Mt Zr(0)-PCH	78	0.002	12	0.051	6.92	0.057
	180	0.008	16	0.088	6.24	0.234
Zr(3)-Mt	231	0.083	103	0.194	3.36	0.326
Zr(3)-PCH	445	0.063	50	0.394	3.53	1.031
Zr(6)-Mt	277	0.088	122	0.324	3.24	0.367
Zr(6)-PCH	950	0.101	142	0.804	3.02	0.993
Zr(12)-Mt	287	0.095	187	0.229	3.19	0.386
Zr(12)-PCH	933	0.105	135	0.818	2.95	0.973
Zr(24)-Mt	318	0.091	174	0.32	3.2	0.394
Zr(24)-PCH	908	0.101	145	0.736	2.84	0.953

SSA corresponds to specific surface area, µpore (micropore) volume, µpore SS corresponds to micropore surface area, TPV corresponds to total pore volume, and APD corresponds to average pore diameter.

**Table 3 materials-15-05567-t003:** Microtextural properties of Zr(6)-PCH precursor calcined at different temperatures.

Samples	SSA (m^2^/g)	µpore Volume (mL/g)	TPV (cm^3^/g)	APD (nm)
Zr(6)-PCH	401	0.067	0.287	2.87
550 °C *	950	0.104	0.801	2.62
650 °C	808	0.082	0.572	2.83
750 °C	783	0.073	0.572	2.83
850 °C	697	0.062	0.460	2.64
900 °C	562	0.000	0.554	3.87

* Calcination temperature, SSA corresponds to specific surface area, µpore volume corresponds to micropore volume, TPV corresponds to total pore volume, and APD corresponds to average pore diameter.

**Table 4 materials-15-05567-t004:** Langmuir model parameters obtained from the fitting of experimental data for different Zr(X)-PCH materials.

Materials	q_max_ (mg/g)	K_L_ (L/mg)	R^2^	*Γ*_max_ * (mg/m^2^)
Ca-Mt	57	0.0289	0.9854	0.736
Zr(3)-PCH	224	0.0397	0.99710	0.505
Zr(6)-PCH	346	0.2124	0.9996	0.364
Zr(12)-PCH	301	0.0671	0.9997	0.322
Zr(24)-PCH	265	0.0408	0.99905	0.292

* *Γ*_max_ corresponds to maximal surface concentrations.

**Table 5 materials-15-05567-t005:** Basic blue-41. Adsorption properties of some selected aluminosilicate materials.

Materials	q_max_ (mg/g)	Reference
Zr(X)-PCH	224–346	This study
Al-PCH material	274	[23]
Al-Pillared clays	88	[23]
Zr- Pillared clays	114	[23]
Nanoporous silica	345	[48]
Natural zeolite	77	[47]
Saudi Local clay	50–70	[46]
Brick waste materials	60–70	[33]
Sodalite zeolite	39	[49]
Zeolite X	27	[49]
bentonite -poly(p-hydroxybenzoic acid) composite	173	[41]
Mn modified diatomite	62	[50]

**Table 6 materials-15-05567-t006:** Estimated volumes and percentage occupation of pores of the maximum removed amounts (q_max_) of basic blue-41 by Zr(X)-PCH materials.

Materials	q_max_ (mg/g)	Estimated Volume (cc/g)	Experimental Percentage Occupation (%)	Calculated Pore Occupation (%)
Zr(3)-PCH	224	0.2335	59.50	63.02
Zr(6)-PCH	346	0.360	44.77	43.02
Zr(12)-PCH	301	0.3135	38.50	45.12
Zr(24)-PCH	265	0.2765	37.56	48.68

**Table 7 materials-15-05567-t007:** Thermodynamic parameters of adsorption of basic-blue-41 using Zr(6)-PCH material.

Parameter	Temperature			
	298 K	303 K	313 K	323 K
ΔG° (kJ mol^−1^)	−3.988	−4.391	−4.878	−5.192
ΔS° (kJ mol^−1^ K)	0.048			
ΔH° (kJ mol^−1^)	10.082			

## Data Availability

Not applicable.

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
