# Peer review of "Enhancement Properties of Zr Modified Porous Clay Heterostructures for Adsorption of Basic-Blue 41 Dye: Equilibrium, Regeneration, and Single Batch Design Adsorber"

_materials, 2022, doi:10.3390/ma15165567_

Round 1

Reviewer 1 Report

Review of paper ‘Enhancement properties of Zr modified porous clay heterostructures for adsorption of Basic-Blue 41 dye: equilibrium, regeneration, and single batch design adsorber’ prepared by Saheed Abiodun Popoola, Hmoud Al Dmour, Souad Al-Rakas, Is Fatimah, Yan Liu, Ahmed Mohmoud, and Fethi Kooli.

The manuscript materials-1851008 is focused on the presentation of new adsorbents for the removal of dye. The influence of many factors, for example concentration, Zr content, and pH, on the efficiency of the recovery process has been studied. Moreover, the regeneration method was also considered. In my opinion, this article is worth publishing in Materials after a minor revision. I have some suggestions that authors may consider prior to publication of this work:

1. The authors wrote that ‘Raw clay (Ca-Mt) was acquired from Purdue University’. University cannot be a source of clay. Provide accurate information in this regard.

2. The article lacks consistency in the writing of temperature unities: once it is with a space and once without (look at lines 160-170, for example). This should be standardized throughout the manuscript.

3. The authors examined a range of variables that affect process performance. I suggest adding information on optimal conditions. At the same time, there is a lack of comparison of the authors' own research with those available in the literature. There are a number of works on dye addition. A comparison of the efficiency of the dye removal process by different sorbents would allow to indicate the advantages and disadvantages of the proposed own solution.

4. In the manuscript, 8 publications of the correspondent author are cited. With a total number of references of 52 is 15%. Self-citation, although sometimes necessary, should not be used too often. Please correct if all references are necessary. In addition, citation 51 should be corrected (numbers 51 to 53 are one publication).

5. English needs correction. For example, in the abstract: ‘This value was dependent on the zirconium content in the PCHs’ or ‘The amount of PCH materials required depended on the target percentage of adsorption at specific volume and the initial concentration of the basic-blue-41 dye solution’.

Author Response

  1. The authors wrote that ‘Raw clay (Ca-Mt) was acquired from Purdue University’. University cannot be a source of clay. Provide accurate information in this regard.

Indeed, the raw clay (Ca-Mt) was purchased from Source Clays Repository based in Purdue University. This information was added in the text as well.

  1. The article lacks consistency in the writing of temperature unities: once it is with a space and once without (look at lines 160-170, for example). This should be standardized throughout the manuscript.

This remark was taken in account and the writing of the temperature unit was consistently rechecked, and no space was left between the value and the unit.

  1. The authors examined a range of variables that affect process performance. I suggest adding information on optimal conditions. At the same time, there is a lack of comparison of the authors' own research with those available in the literature. There are a number of works on dye addition. A comparison of the efficiency of the dye removal process by different sorbents would allow to indicate the advantages and disadvantages of the proposed own solution.

The optimal conditions to adsorb the basic-blue 41 was added in paragraph 3.2.6.

The reviewer indicated that there are a number of works on dye addition. The authors agreed with the reviewer, and the last paragraph is section 3.2.5 was modified.

A new table where a collection of adsorbents used in the literature, for the adsorption of BB-41 is reported in Table 5. Only the silica and aluminosilicates were mentioned in this present study, due to the similarity of these materials to our prepared samples.

New references were added in the Table and  the order was changed accordingly in the text

  1. In the manuscript, 8 publications of the correspondent author are cited. With a total number of references of 52 is 15%. Self-citation, although sometimes necessary, should not be used too often. Please correct if all references are necessary. In addition, citation 51 should be corrected (numbers 51 to 53 are one publication).

The authors have tried to avoid the self publication, and one reference was deleted [24].The others were necessary to support the results

The number of references was rechecked, and it was changed.

  1. English needs correction. For example, in the abstract: ‘This value was dependent on the zirconium content in the PCHs’ or ‘The amount of PCH materials required depended on the target percentage of adsorption at specific volume and the initial concentration of the basic-blue-41 dye solution’.

The authors have done their efforts to check the manuscript

Reviewer 2 Report

In this article, Zirconium porous clay heterostructures (Zr-PCH) were synthesized using intercalated clay minerals by zirconium species with different contents of zirconium. The presence of zirconium and silica species was confirmed by X-ray diffraction, x-ray fluorescence, and solid nuclear magnetic resonance. The results of this paper are new and recommended to be published, but some small details need to be corrected and improved.

1The picture of the manuscript is suggested to be readjusted. Such as Figures 2 and 3.

2The paper should further illustrate or contrast the innovation.

(3)  A manuscript should be neatly arranged before submission. As shown in Formulas (1) (2)

 (4)  It is suggested to add some literature to enhance the background of this paper, such as “Amplitude and frequency tunable absorber in the terahertz range”, Results in Physics, 2022, 34:105263

“Analyzing broadband tunable metamaterial absorber by using symmetry model”, Optics Express, 2021, 29(25): 41475-41484

Author Response

The authors would like to thank the reviewer for his comments, te authors did all their bests to consider them

In this article, Zirconium porous clay heterostructures (Zr-PCH) were synthesized using intercalated clay minerals by zirconium species with different contents of zirconium. The presence of zirconium and silica species was confirmed by X-ray diffraction, x-ray fluorescence, and solid nuclear magnetic resonance. The results of this paper are new and recommended to be published, but some small details need to be corrected and improved.

(1)The picture of the manuscript is suggested to be readjusted. Such as Figures 2 and 3

The reviewer suggested to readjust the figures 2 and 3. I guess the editing team during the final draft will take care of them.

(2)The paper should further illustrate or contrast the innovation.

The innovation of this paper is to modify the properties of Zr-PCH materials by direct insertion of different amounts of Zr in the silica framework using intercalated clay precursors with Zr species. This method has helped to reduce the chemical waste and the steps during the synthesis process. This statement was added in lines 93-96.

 (3)  A manuscript should be neatly arranged before submission. As shown in Formulas (1) (2)

The authors apologized for the inconvenience, this was related using different versions of the word software and the windows version.

 (4)  It is suggested to add some literature to enhance the background of this paper, such as “Amplitude and frequency tunable absorber in the terahertz range”, Results in Physics, 2022, 34:105263

 “Analyzing broadband tunable metamaterial absorber by using symmetry model”, Optics Express, 2021, 29(25): 41475-41484

In the introduction, the authors have been interested to a specific family of adsorbents which is the porous clay heterostructures based on clay minerals and their modification. Of course, there are many different types of adsorbers. The materials mentioned by the reviewer are interesting and their modification by physical methods seems very interesting. The authors hope to take in account this method and mention it in their next paper.

Reviewer 3 Report

This article reports on the adsorption removal of Basic-Blue 41 dye using Zr modified porous clay hetero- structures as adsorbent. The study is interesting from environmental point of view especially water purification technology. However, the study is incomplete and needs further work as pointed out in the following comments.

Particular comments:

1.     Thermodynamics plays main role in surface science and the adsorption phenomenon is usually explained by thermodynamic parameters like Gibb energy, enthalpy and entropy changes. These parameters are necessary to differentiate between spontaneity , non-spontaneity, exothermic and endothermic behavior of adsorption. But the authors have completely neglected thermodynamic study in the present work.

2.     They studied the adsorption of model sample of Basic blue -41 dye but did not test the adsorption behavior of Zr modified porous clay hetero- structures towards BB-41 dye containing real samples . Industrial effluents contain several co-existing ions in addition to dyes, but in this study effects of such ions on adsorption are not included.

3.     Investigation of the kinetic aspects of the adsorption process would make the article more interesting for reader.

General comments

4.     Replace gram with g in the abstract and throughout the text.

5.     Replace hybrid precursor with hybrid precursor in the introduction.

6.     Taking in account the pore size..... Should be written as taking in to account the pore size........

7.     Add " been"  after have in the sentence " In water  purification, they have used as adsorbents  for ......

8.     The abbreviation (MAS-NMR) for solid nuclear magnetic resonance seems inappropriate.

9.     What do the authors mean from " free pH sealed tubes".

10.  Ultra vilote -Visible spectrophotomer is usualy written as UV-Visible rather than U-Visible .....

11.  Please also provide PXRD of Zr in Figure 1.

12.   They write that preliminary studies have indicated that the Langmuir model was the most adequate to estimate the maximum removal capacity loaded on the surface of the solid.

It is not clear whether they have also tried other isothers or not. If the answer is yes then they must provide data for other isotherms such as Tmpkin and D-R isotherms.

13.   They must compare their data with literature for justifying their claims.

14.  There are several poor sentences in the manuscript. Revision is suggested.

Author Response

The authors would like to thank the reviewer for his comments, te authors did all their bests to consider them

This article reports on the adsorption removal of Basic-Blue 41 dye using Zr modified porous clay hetero- structures as adsorbent. The study is interesting from environmental point of view especially water purification technology. However, the study is incomplete and needs further work as pointed out in the following comments.

Particular comments:

  1. Thermodynamics plays main role in surface science and the adsorption phenomenon is usually explained by thermodynamic parameters like Gibb energy, enthalpy and entropy changes. These parameters are necessary to differentiate between spontaneity , non-spontaneity, exothermic and endothermic behavior of adsorption. But the authors have completely neglected thermodynamic study in the present work.

The effect of temperature on the adsorption of basic blue-41 using Zr(6)-PCH material was added in the text. New paragraph was added ( 3.2.7. Adsorption thermodynamic).  The authors have studied the adsorption of Basic Blue-41 at different temperatures of 25, 30, 40 and 50 oC, using an initial concentration of 200 mg/L and 0.050 g of Zr(6)-PCH and a volume of 50 mL.

  1. They studied the adsorption of model sample of Basic blue -41 dye but did not test the adsorption behavior of Zr modified porous clay hetero- structures towards BB-41 dye containing real samples . Industrial effluents contain several co-existing ions in addition to dyes, but in this study effects of such ions on adsorption are not included.

The authors have performed the adsorption of BB-41 dye from a model solution, but they did not test the adsorption behaviour on real samples.

This study is considered as preliminary, and a complete study will be performed in the near future including the effect of co-existing ions.

  1. Investigation of the kinetic aspects of the adsorption process would make the article more interesting for reader.

As the authors have mentioned above, the complete study will be performed in the near future.

General comments

  1. Replace gram with g in the abstract and throughout the text.

The authors have made the necessary changes as requested.

  1. Replace hybrid precursor with hybrid precursor in the introduction.

The comment was taken in account and it was changed.

  1. Taking in account the pore size..... Should be written as taking in to account the pore size........

The authors have corrected the statement accordingly.

  1. Add " been" after have in the sentence " In water  purification, they have used as adsorbents  for ......

The authors have add “been” in the adequate place.

  1. The abbreviation (MAS-NMR) for solid nuclear magnetic resonance seems inappropriate.

The authors have checked the abbreviation of solid nuclear  magnetic resonance , the full name is magic-angle spinning nuclear magnetic resonance, and the abbreviation now is correct. The necessary modifications were performed in the text.

  1. What do the authors mean from " free pH sealed tubes".

This means that the experiments were performed without pH adjustment, means natural pH.

  1. Ultra vilote -Visible spectrophotomer is usually written as UV-Visible rather than U-Visible It was rechecked in the text and the authors have made the change
  2. Please also provide PXRD of Zr in Figure 1.

The reviewer has suggested to add the PXRD pattern of Zr, I guess it is Zr metal, in this case it will not add any information, because the Zr species could be as Zirconium oxide. The PXRD pattern of ZrO2 phase was added in Figure 3, and no reflection was detected , and it could  be incorporated in the silica matrix, ( see lines 222-224)

  1. They write that preliminary studies have indicated that the Langmuir model was the most adequate to estimate the maximum removal capacity loaded on the surface of the solid. The Langmuir model

The authors apologised for this misunderstand, the authors meant by  “preliminary”  “previous”, and indeed, the Langmuir model was the often used model to estimate the maximum removal capacity loaded on the surface of the solid. Please see paragraph 3.2.5.

It is not clear whether they have also tried other isotherms or not. If the answer is yes then they must provide data for other isotherms such as Tampkin and D-R isotherms.

The authors did not try other models, because they were looking for maximum adsorption capacity loaded on the surface of the solid.

However, it will be done in near future to complete the whole story

  1. They must compare their data with literature for justifying their claims.

New table (Table 5) was added and the data were reported with some comments ( lines 418-423)

  1. There are several poor sentences in the manuscript. Revision is suggested.

The authors have revised the manuscript and tried to improve some sentences

Round 2

Reviewer 3 Report

The authors have provided revised version of their manuscript for further review. I would like to ask them to address my earlier comments 2 and 3 . This will not on;y improve the quality of the manuscript but will also make it very interesting for readers.

The comments are

1. They studied the adsorption of model sample of Basic blue -41 dye but did not test the adsorption behavior of Zr modified porous clay hetero- structures towards BB-41 dye containing real samples . Industrial effluents contain several co-existing ions in addition to dyes, but in this study effects of such ions on adsorption are not included.

2. They should also focus on the kinetic aspects of adsorption of the dye on their material,

Author Response

The authors would like to thank the reviewer for his comments, te authors did all their bests to consider them

This article reports on the adsorption removal of Basic-Blue 41 dye using Zr modified porous clay hetero- structures as adsorbent. The study is interesting from environmental point of view especially water purification technology. However, the study is incomplete and needs further work as pointed out in the following comments.

Particular comments:

  1. Thermodynamics plays main role in surface science and the adsorption phenomenon is usually explained by thermodynamic parameters like Gibb energy, enthalpy and entropy changes. These parameters are necessary to differentiate between spontaneity , non-spontaneity, exothermic and endothermic behavior of adsorption. But the authors have completely neglected thermodynamic study in the present work.

The effect of temperature on the adsorption of basic blue-41 using Zr(6)-PCH material was added in the text. New paragraph was added ( 3.2.7. Adsorption thermodynamic).  The authors have studied the adsorption of Basic Blue-41 at different temperatures of 25, 30, 40 and 50 oC, using an initial concentration of 200 mg/L and 0.050 g of Zr(6)-PCH and a volume of 50 mL.

  1. They studied the adsorption of model sample of Basic blue -41 dye but did not test the adsorption behavior of Zr modified porous clay hetero- structures towards BB-41 dye containing real samples . Industrial effluents contain several co-existing ions in addition to dyes, but in this study effects of such ions on adsorption are not included.

The authors have performed the adsorption of BB-41 dye from a model solution, but they did not test the adsorption behaviour on real samples.

This study is considered as preliminary, and a complete study will be performed in the near future including the effect of co-existing ions.

  1. Investigation of the kinetic aspects of the adsorption process would make the article more interesting for reader.

As the authors have mentioned above, the complete study will be performed in the near future.

General comments

  1. Replace gram with g in the abstract and throughout the text.

The authors have made the necessary changes as requested.

  1. Replace hybrid precursor with hybrid precursor in the introduction.

The comment was taken in account and it was changed.

  1. Taking in account the pore size..... Should be written as taking in to account the pore size........

The authors have corrected the statement accordingly.

  1. Add " been" after have in the sentence " In water  purification, they have used as adsorbents  for ......

The authors have add “been” in the adequate place.

  1. The abbreviation (MAS-NMR) for solid nuclear magnetic resonance seems inappropriate.

The authors have checked the abbreviation of solid nuclear  magnetic resonance , the full name is magic-angle spinning nuclear magnetic resonance, and the abbreviation now is correct. The necessary modifications were performed in the text.

  1. What do the authors mean from " free pH sealed tubes".

This means that the experiments were performed without pH adjustment, means natural pH.

  1. Ultra vilote -Visible spectrophotomer is usually written as UV-Visible rather than U-Visible It was rechecked in the text and the authors have made the change
  2. Please also provide PXRD of Zr in Figure 1.

The reviewer has suggested to add the PXRD pattern of Zr, I guess it is Zr metal, in this case it will not add any information, because the Zr species could be as Zirconium oxide. The PXRD pattern of ZrO2 phase was added in Figure 3, and no reflection was detected , and it could  be incorporated in the silica matrix, ( see lines 222-224)

  1. They write that preliminary studies have indicated that the Langmuir model was the most adequate to estimate the maximum removal capacity loaded on the surface of the solid. The Langmuir model

The authors apologised for this misunderstand, the authors meant by  “preliminary”  “previous”, and indeed, the Langmuir model was the often used model to estimate the maximum removal capacity loaded on the surface of the solid. Please see paragraph 3.2.5.

It is not clear whether they have also tried other isotherms or not. If the answer is yes then they must provide data for other isotherms such as Tampkin and D-R isotherms.

The authors did not try other models, because they were looking for maximum adsorption capacity loaded on the surface of the solid.

However, it will be done in near future to complete the whole story

  1. They must compare their data with literature for justifying their claims.

New table (Table 5) was added and the data were reported with some comments ( lines 418-423)